# Hierarchical Optimal Transport for Comparing Histopathology Datasets

**Anna Yeaton**[1]                                                                                    AY1392@NYU.EDU
**Rahul G. Krishnan**[4]                                                                  RAHULGK@CS.TORONTO.EDU
**Rebecca Mieloszyk**[2]                                              REBECCA.MIELOSZYK@MICROSOFT.COM
**David Alvarez-Melis**[3]                                               ALVAREZ.MELIS@MICROSOFT.COM
**Grace Huynh**[2]                                                               GRACE.HUYNH@MICROSOFT.COM

[1] *Department of Pathology, NYU Grossman School of Medicine*

[4] *Department of Computer Science, University of Toronto*

[2] *Microsoft Research, Redmond*

[3] *Microsoft Research, New England*

**Editors:** Under Review for MIDL 2022

## Abstract

Scarcity of labeled histopathology data limits the applicability of deep learning methods to under-profiled cancer types and labels. Transfer learning allows researchers to overcome the limitations of small datasets by pre-training machine learning models on larger datasets *similar* to the small target dataset. However, similarity between datasets is often determined heuristically. In this paper, we propose a principled notion of distance between histopathology datasets based on a hierarchical generalization of optimal transport distances. Our method does not require any training, is agnostic to model type, and preserves much of the hierarchical structure in histopathology datasets imposed by tiling. We apply our method to H&E stained slides from The Cancer Genome Atlas from six different cancer types. We show that our method outperforms a baseline distance in a cancer-type prediction task. Our results also show that our optimal transport distance predicts difficulty of transferability in a tumor vs. normal prediction setting.

**Keywords:** Histopathology, domain adaptation, transfer learning, optimal transport

## 1. Introduction

Histopathology images are routinely used in the diagnostic workup of many cancers. Beyond the standard identification of tumor grade and subtype, histopathology images also contain an abundance of visual information that may have predictive and prognostic value. Recent advances in deep learning for histopathology are facilitating the extraction of this information, enabling prediction of genetic alterations, treatment response and survival (Coudray et al., 2018; Muhammad et al., 2021; Wulczyn et al., 2021; Echle et al., 2020).

Despite the promise of supervised deep learning, the large, labeled datasets required to train complex networks on histopathology images are scarce, especially in less common cancers and label types. To overcome data scarcity during model training, transfer learning techniques can be utilized by pre-training models on a larger—ideally similar—dataset. The model can then be fine-tuned on the small target dataset of interest. Until now, determining

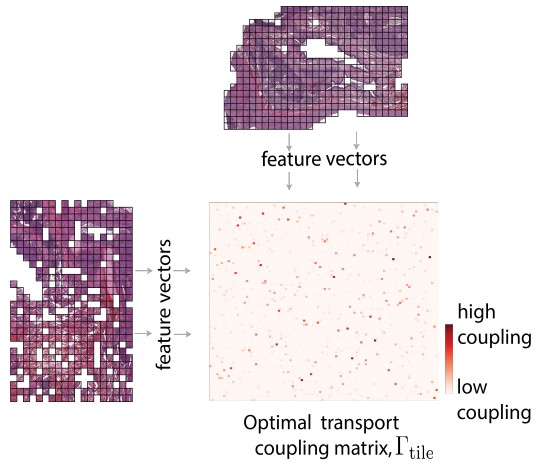

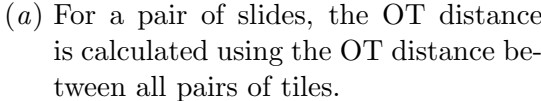

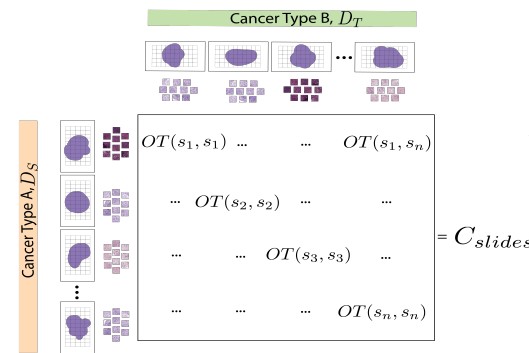

(*a*) For a pair of slides, the OT distance is calculated using the OT distance between all pairs of tiles.

(*b*) For a pair of datasets, the OT distance is calculated using the OT distance between all pairs of slides

Figure 1: Schematic of how HHOT distances are calculated at the slide (a) and dataset (b) level. Tiling of individual slides, done to overcome memory limits, introduces hierarchical structure to the calculation of OT between different datasets.

which similar dataset to use for pre-training in histopathology has been driven by intuition or trial and error (Srinidhi et al., 2021).

In this paper, we introduce a principled approach to measuring similarities between histopathology datasets. Specifically, we propose a novel distance between histopathology datasets which we call Hierarchical Histopathology Optimal Transport (HHOT) based on Optimal Transport (OT), a method to compare distributions. Our method uses OT to compute the distance between individual slides and to compute the distances between entire datasets (Figure 1). We show that HHOT is highly predictive of transfer learning accuracy in a tumor vs. normal prediction setting, and that it is significantly faster than a naive (non-hierarchical) Optimal Transport approach.

## 2. Background

Optimal Transport (OT) is a mathematical framework centered around the goal of comparing probability distributions, with deep theory (Villani, 2003, 2008; Santambrogio, 2015) and applications to various fields, ranging from economics (Galichon, 2016) to meteorology (Cullen and Maroofi, 2003). Although it can be formulated in more general settings, in this work we are interested in its discrete Euclidean formulation, which considers two finite collections of points $\{\mathbf{x}^{(i)}\}_{i=1}^n$, $\{\mathbf{y}^{(j)}\}_{j=1}^m$, $\mathbf{x}^{(i)}, \mathbf{y}^{(j)} \in \mathbb{R}^d$, represented as empirical distributions: $\mu = \sum_{i=1}^n \mathbf{p}_i \delta_{\mathbf{x}^{(i)}}$, $\nu = \sum_{j=1}^m \mathbf{q}_j \delta_{\mathbf{y}^{(j)}}$, where $\mathbf{p}$ and $\mathbf{q}$ are probability vectors (non-negative and adding up to one).

At a high level, the goal of OT is to find an optimal correspondence between these distributions, and in doing so, define a notion of similarity between them. Given a cost function (often called the *ground metric*) between pairs of points $c : \mathbb{R}^d \times \mathbb{R}^d \to \mathbb{R}+$, the

goal of OT is to find a correspondence between $\mu$ and $\nu$ with minimal cost. Formally, the Kantorovich formulation of discrete OT seeks a coupling matrix $\Gamma \in \mathbb{R}^{n \times m}$ that solves:

$$\mathrm{OT}_c(\mu, \nu) \overset{\text{def.}}{=} \min_{\Gamma \in \Pi(\mu, \nu)} \langle \Gamma, \mathbf{C} \rangle = \sum_{ij} \Gamma_{ij} \mathbf{C}_{ij}, \tag{1}$$

where $\mathbf{C}_{ij} \overset{\text{def.}}{=} c(\mathbf{x}^{(i)}, \mathbf{y}^{(j)})$. The constraint set $\Pi(\mu, \nu)$ enforces $\Gamma$ to be *measure-preserving*, i.e., to have $\mu$ and $\nu$ as its marginals:

$$\Pi(\mu, \nu) \overset{\text{def.}}{=} \{\Gamma \in \mathbb{R}_+^{n \times m} \mid \Gamma \mathbf{1} = \mathbf{p},\ \Gamma^\top \mathbf{1} = \mathbf{q}\}. \tag{2}$$

It can be shown that for $c(\mathbf{x}, \mathbf{y}) = \|\mathbf{x} - \mathbf{y}\|^p$, $\mathrm{OT}_c(\mu, \nu)^{1/p}$ is a proper distance metric between distributions (i.e., satisfies all metric axioms) (Peyré and Cuturi, 2019). As noted above, the coupling matrix $\Gamma$ can be interpreted as a soft matching or probabilistic correspondence between the elements of $\mu$ and $\nu$, in the sense that $\Gamma_{ij}$ is high if $\mathbf{x}^{(i)}, \mathbf{y}^{(j)}$ are in 'correspondence', and low otherwise. In some cases (such as the common case $n = m$ and $\mathbf{p}, \mathbf{q}$ uniform), the optimal coupling turns out to be sparse, in which case $\Gamma$ defines a deterministic one-to-one mapping between the $\mathbf{x}^{(i)}$'s and $\mathbf{y}^{(j)}$'s.

Problem (1) is a linear programming problem and thus solvable exactly in $O(n^3)$ time (Peyré and Cuturi, 2019). This makes it impractical even for moderately sized collections of points. However, seminal work by Cuturi (2013) showed that a regularized version of this problem can be solved much more efficiently. The regularization consists of adding a entropy term on the objective:

$$\mathrm{OT}_{c,\varepsilon}(\mu, \nu) \overset{\text{def.}}{=} \min_{\Gamma \in \Pi(\mu, \nu)} \langle \Gamma, \mathbf{C} \rangle + \varepsilon \mathrm{H}(\Gamma). \tag{3}$$

This *entropy-regularized* OT problem can be solved very efficiently using the Sinkhorn-Knopp algorithm (Cuturi, 2013). One downside of this regularization is that $\mathrm{OT}_{c,\varepsilon}(\mu, \mu) \neq 0$, which in particular implies this quantity is no longer a valid distance. To alleviate this, prior work (Genevay et al., 2018; Salimans et al., 2018) has considered a *debiased* version this quantity, also known as the Sinkhorn divergence:

$$\mathrm{SD}_{c,\varepsilon}(\mu, \nu) \overset{\text{def.}}{=} \mathrm{OT}_{c,\varepsilon}(\mu, \nu) - \tfrac{1}{2}\big(\mathrm{OT}_{c,\varepsilon}(\mu, \mu) + \mathrm{OT}_{c,\varepsilon}(\nu, \nu)\big) \tag{4}$$

In addition to satisfying $\mathrm{SD}_{c,\varepsilon}(\mu, \nu) \geq 0$, with equality if and only if $\mu = \nu$, this divergence comes with many other desirable theoretical properties: it is positive, convex, metrizes weak converge in distribution (Feydy et al., 2019), and leads to faster statistical rates of estimation of the exact OT problem (Chizat et al., 2020).

In Section 4.1, we will introduce our method using $\mathrm{OT}_{c,\varepsilon}(\mu, \nu)$ for notational simplicity, noting that it can naturally use $\mathrm{SD}_{c,\varepsilon}(\mu, \nu)$ instead. In addition, we will drop $c$ and $\epsilon$ from the notation for OT when these are clear from the context.

## 3. Related work

Our HHOT method for histopathology builds on previous work in hierarchical OT for other domains and previous work in non-hierarchical OT within histopathology. For example, Yurochkin et al. (2019) have described a hierarchical OT method for measuring distances

between documents, using words and topics as the hierarchical levels. In addition, specifically for histopathology images, non-hierarchical OT has been used to compare individual cell morphology (Basu et al., 2014; Wang et al., 2011) and to quantify domain shift at the tile level (Stacke et al., 2021). In addition, the relationship between OT-calculated dataset distances and the difficulty of transferability has been previously described by (Alvarez-Melis and Fusi, 2020; Gao and Chaudhari, 2021; Achille et al., 2021), although the notion of distance they define is generic and thus does not leverage the hierarchical nature of individual datasets.

## 4. Optimal transport between histopathology datasets

### 4.1. Hierarchical Histopathology Optimal Transport

We consider a pair of histopathology datasets, $\mathsf{D}_a$ and $\mathsf{D}_b$, collected from different tissues, centers, or populations. We seek a notion of distance that lets us assess their similarity. Each dataset consists of slides $\mathbf{s}$, which in turn are subdivided into tiles $\mathbf{t}$. Let $n$ and $m$ denote, respectively, the number of slides in each dataset. In addition, we denote by $n_i$ the number of tiles in the $i$-th slide of the first dataset, and analogously for $m_j$.

We can view $\mathsf{D}_a$ and $\mathsf{D}_b$ as point clouds or, more formally, empirical distributions as described in section 2. From this viewpoint, we can think of these datasets as samples of slide images sampled from two different underlying distributions. Unless additional information is provided, we can simply take the weights associated to each slide ($\mathbf{p}$ and $\mathbf{q}$) to be uniform, as is typically done in practical application of OT to point clouds or images (Peyré and Cuturi, 2019). After defining a suitable notion of distance between pairs of slides, one could in principle use problem (3) (or its debiased counterpart, eq. (4)) to obtain a notion of similarity between $\mathsf{D}_a$ and $\mathsf{D}_b$. However, this naive approach would require loading entire slides into memory, which is computationally infeasible—precisely the reason why these images are tiled in the first place.

Our proposed solution to this computational hurdle is to interpret the slides themselves as collections (formally, distributions) of tiles, and use OT once more, now to define a notion of distance between these. To this end, we first define the cost between individual tiles as the distance between them. Although we could in principle use the Euclidean distance between their raw pixel representations, a more meaningful comparison can be obtained by first embedding these images in some lower dimensional space (e.g., using a neural network pre-trained on a large image dataset), and then computing a distance between them. Hence, we define $c(\mathbf{t}_u, \mathbf{t}_v) = \|\phi(\mathbf{t}_u) - \phi(\mathbf{t}_v)\|^2$, where $\phi$ is a pre-trained encoder.

We collect all such pairwise costs in a matrix $\mathbf{C}_{\text{tile}}$ of size $n_i \times m_j$, and solve the corresponding OT problem:

$$\text{OT}_{\phi,\varepsilon}(\mathbf{s}_i, \mathbf{s}_j) = \min_{\Gamma \in \Pi(\mathbf{p}_i, \mathbf{q}_j)} \langle \Gamma_{\text{tile}}, \mathbf{C}_{\text{tile}}^{ij} \rangle + \varepsilon \text{H}(\Gamma). \tag{5}$$

As in section 2, we can also define a debiased version of this slide-to-slide distance:

$$\text{SD}_{\phi,\varepsilon}(\mathbf{s}_i, \mathbf{s}_j) = \text{OT}_{\phi,\varepsilon}(\mathbf{s}_i, \mathbf{s}_j) - \tfrac{1}{2}\big(\text{OT}_{\phi,\varepsilon}(\mathbf{s}_i, \mathbf{s}_i) + \text{OT}_{\phi,\varepsilon}(\mathbf{s}_j, \mathbf{s}_j)\big) \tag{6}$$

Compared to other possible ways to compare slides using their tiles (like using a mean or centroid tile), this OT-based approach is appealing because (i) it does not lose information

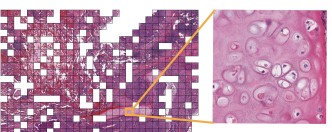 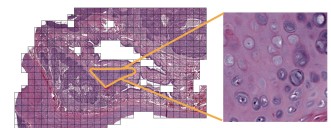 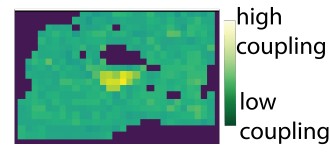

($a$) Source tile visualizing cartilage within a LUAD slide.

($b$) Target slide of LUAD. The orange highlighted region is cartilaginous.

($c$) Heatmap of coupling between the source tile and target slide. High coupling is observed in the cartilaginous region.

Figure 2: Representative example of a source tile tightly coupling to target tiles of the same tissue type, suggesting that HHOT between tiles implicitly incorporates biological information.

by aggregating the tiles, and (ii) it recovers some of the *global* structure that it lost by tiling, i.e., the relation between the tiles in the context of the slide. Specifically, in operationalizing similarity through matching, problem (5) seeks to find corresponding tiles across slides, and does it coherently as a result of the marginal constraint (e.g., a single tile cannot be matched to all tiles in the other slide).

Once we have computed (5) for every pair of slides $(\mathbf{s}_i, \mathbf{s}_j)$ from the two datasets, we collect them in a matrix $\mathbf{C}_{\text{slide}}$ with entries $[\mathbf{C}_{\text{slide}}]_{ij} = \text{OT}_\varepsilon(\mathbf{s}_i, \mathbf{s}_j)$. With this, finally we have a *ground cost* between slides, which we can use to compute the sought-after distance between datasets using OT once more:

$$\text{OT}_\varepsilon(\mathsf{D}_a, \mathsf{D}_b) = \min_{\Gamma \in \Pi(\mathbf{p}, \mathbf{q})} \langle \Gamma, \mathbf{C}_{\text{slide}} \rangle + \varepsilon H(\Gamma), \tag{7}$$

and its debiased counterpart:

$$\text{SD}_{\phi,\varepsilon}(\mathsf{D}_a, \mathsf{D}_b) = \text{OT}_{\phi,\varepsilon}(\mathsf{D}_a, \mathsf{D}_b) - \tfrac{1}{2}\big(\text{OT}_{\phi,\varepsilon}(\mathsf{D}_a, \mathsf{D}_b) + \text{OT}_{\phi,\varepsilon}(\mathsf{D}_a, \mathsf{D}_b)\big). \tag{8}$$

### 4.2. Computational Implementation

We use the python optimal transport (POT) (Flamary et al., 2021) and geomloss (Feydy et al., 2019) libraries to solve the individual OT problems (5) and (7). Using the vanilla Sinkhorn algorithm, solving the first of these to $\delta_1$-accuracy has $O(n_i m_j / \delta_1)$ computational complexity (Altschuler et al., 2017), and analogously $O(nm/\delta_2)$ for the latter. Taking $\delta_1 = \delta_2$, the total complexity scales as $O((nm + \sum_{ij} n_i m_j)/\delta)$. Figure 5($c$) shows empirical runtimes for our method. Our implementation of HHOT can be found here: `https://github.com/ayeaton/HHOT`

## 5. Methodology and Results

We used whole slide images retrieved from the TCGA (https://portal.gdc.cancer.gov/) for six common cancer types, including both primary tumor samples and matched normal

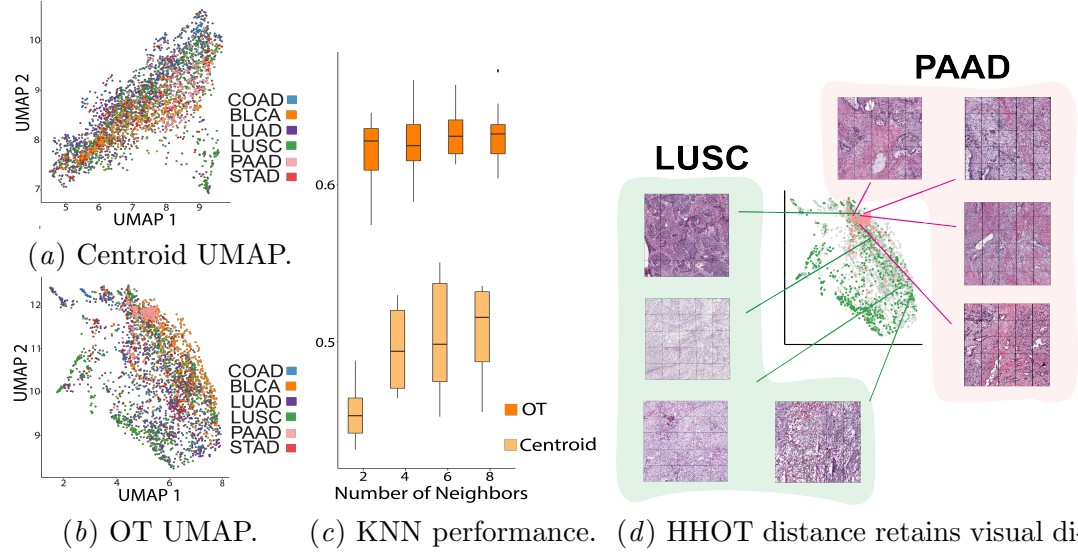

(a) Centroid UMAP.  (b) OT UMAP.  (c) KNN performance.  (d) HHOT distance retains visual diversity within cancer datasets.

Figure 3: HHOT enables better clustering of slide images by cancer type, as compared to the centroid tile distance method. HHOT also reflects the diversity of images within a single cancer type. This suggests that HHOT distances better preserve morphological information.

samples. Specifically we focused on Stomach Adenocarcinoma (STAD), Bladder Urothelial Carcinoma (BLCA), Lung Adenocarcinoma (LUAD), Lung Squamous-cell Carcinoma (LUSC), Colon Adenocarcinoma (COAD), and Pancreatic Adenocarcinoma (PAAD). Slide images were tiled into 512x512 pixel non-overlapping images at 20x magnification, and tiles with more than 50 percent background were discarded as described in (Coudray et al., 2018; Noorbakhsh et al., 2020). We extracted a median of 474 tiles for BLCA, 777 for COAD, 445 for LUAD, 448 for LUSC, 278 for PAAD and 622 for STAD. We then used Inception-V3 pre-trained on ImageNet to compress 512x512 pixel images to 2048 length vectors. For comparison of collections of tiles, we set the regularization parameter $\varepsilon$ to 0.25.

We calculated OT between slides, and found that tiles with the highest coupling were those that were visually similar. As an example, we show the OT coupling matrix between tiles from two representative slides in Figure 2. We chose an example tile from the source slide which displays cartilage tissue. We then show the coupling of this source tile to all tiles in the target slide. We visualize the strength of coupling in Figure 2(c); the highest coupling to the cartilage source tile is tightly localized and corresponds to a cartilaginous region in the target. These results demonstrate that an OT solution implicitly incorporates biological structure.

In slide-to-slide comparisons, HHOT preserves more relevant histomorphological information than other methods, such as using a mean or centroid tile. We demonstrate this with a task for clustering the slides by cancer type. We created a matrix $\mathbf{C}_{\text{slide}}$ with entries $[\mathbf{C}_{\text{slide}}]_{ij} = \text{OT}_{\varepsilon}(\mathbf{s}_i, \mathbf{s}_j)$ as described above including all pairs of slides. For reference, we also calculated a distance matrix between slides using the centroid tile of a slide as described in

Howard et al. (2021). We visualized the relationship between slides using UMAP for both OT and centroid-tile distance (Figure 3(a), Figure 3(b)). Visually, we observe that HHOT distance enables better clustering of the slide images by cancer type. Quantitatively, using K-nearest neighbors over K ranging from two to eight, we performed a cancer-type classification task with our two distance matrices as input. We show that OT retains similarity within cancer-types and dissimilarity across cancer-types as expected, and does so better than the centroid-tile distance baseline. Figure 3(d) shows representative examples of how HHOT distances reflect the diversity of images with a single cancer type. HHOT tightly groups PAAD slides, and these slide images are visually very similar. In contrast, HHOT shows more variability in the LUSC slides, and these images are more variable.

We observed that target datasets that are similar to the source dataset based on HHOT distance are more improved by pre-training, i.e. a negative correlation between HHOT distance and transferability. In this paper, we focus on a tumor vs. normal task. For our pre-training tasks, we standardized the total dataset size to 209 slides, and used a cross validation scheme to create four datasets per cancer type with 169 slides for training and 40 for validation. For our fine-tuning task, we standardized the dataset to 65 slides, and used a cross validation scheme to create four datasets per cancer type with 25 slides for training and 40 for validation. We conducted our task over all the pre-training datasets and all the fine-tuning datasets for 16 experiments per cancer type comparison. For each dataset pair, we pre-trained a single-layer perceptron to predict tumor or normal status, using feature vectors $\phi(\mathbf{t})$ and a learning rate of 1e-2. We then fine-tuned this pre-trained model for each of the other cancer types, using only 25 target slides and a learning rate of 1e-10.

We quantify transferability across tasks using the relative improvement in AUC obtained by transfer learning: $\mathsf{Transferability}(D_T, D_S) = (\mathsf{AUC}(D_S \rightarrow D_T) - \mathsf{AUC}(D_T))/\mathsf{AUC}(D_T)$. We observed a negative correlation between the HHOT distance and transferability most strongly for PAAD, LUAD, LUSC, and STAD (Figure 5(a)). Consistently, visually similar datasets, such as the two lung cancer datasets, show the smallest HHOT distance and the highest transferability (green and purple squares and plus in Figure 5(a)). For BLCA and COAD, we observe that the AUC of a model without pre-training is already very high (Figure 5(b)); thus, pre-training on other datasets cannot improve on this already high AUC. This is explicitly quantified for these two cancers by the nearly zero slope of the best fit line (Figure 5(b), boxplot of no-pretaining AUCs aggregating four datasets of 25 target slides).

Finally, we show that not only does HHOT preserve the hierarchical structure of the data and correlate with difficulty of transfer learning, it is also much faster than a flat (non-hierarchical) OT approach. We compare the time it took to compute OT distance between four to eighteen slides, with 100 tiles each and observed that the time it took to calculate OT distance between slides increased in a polynomial order in the flat case (Figure 5(c)).

## 6. Discussion

In this paper we introduced a principled approach to compare histopathology datasets called HHOT. Our work adds to the OT and histopathology literature by proposing a method to compare datasets while also preserving the structure lost by standard tiling approaches. Specifically, we propose to solve first an inner tile-to-tile OT problem for all

Figure 4: HHOT is predictive of model transferability across cancer types. Datasets with large HHOT distances, representing more visually different data, have worse transferability. The regression coefficients are $[\beta, \beta_0]$ $[0.0024, -5.47 \times 10^{-5}]$ (COAD), $[0.0016, -6 \times 10^{-5}]$ (BLCA), $[0.022, -6.8 \times 10^{-4}]$ (PAAD), $[0.042, -1.7 \times 10^{-3}]$ (LUAD), $[0.043, -2 \times 10^{-3}]$ (LUSC), $[0.049, -2.7 \times 10^{-3}]$ (STAD). For COAD and BLCA, the baseline AUC is already quite high (b), so transfer learning with other cancer types results in little improvement. HHOT is faster than non-hierarchical OT between slides (c).

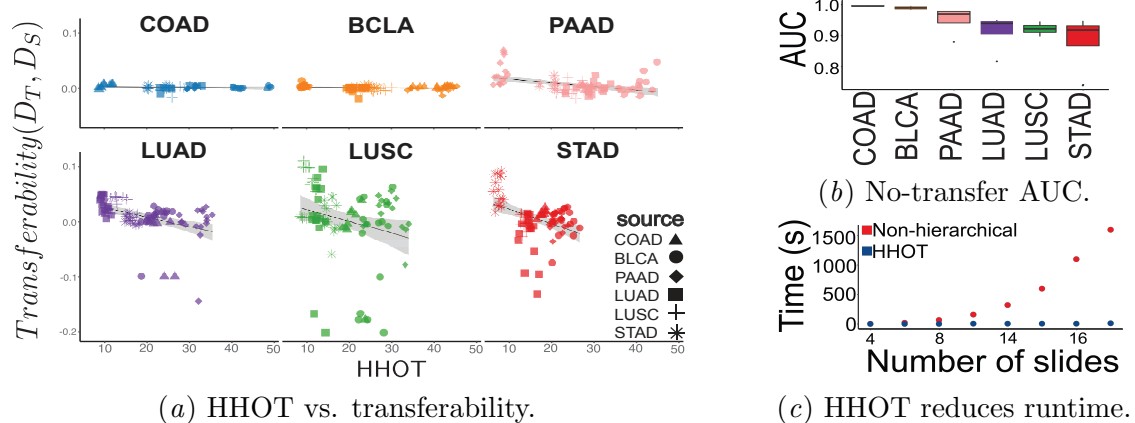

(a) HHOT vs. transferability.

(b) No-transfer AUC.

(c) HHOT reduces runtime.

pairs of slides, and then solve the outer slide-to-slide OT problem between datasets. We first show that correspondences in the OT coupling matrix $\Gamma_{tile}$ map the same type of tissue across slides. We then show that OT distance performs better than a naive, centroid-tile distance based method in a cancer-type prediction task. We also show that HHOT distance between histopathology datasets correlates with transferability. Furthermore, we find that the degree of correlation between HHOT distance and transferability is associated with baseline AUC. Simply, if the task is already close to optimal, it is difficult to improve using pre-training. Finally, we show that HHOT distance is much faster than a naive, flat approach to comparing histopathology datasets. In addition to applications presented in this paper, promising applications of HHOT include outlier detection, clustering analysis, dataset visualization, and facilitating multi-modal integration of whole slide images and molecular data.

In conclusion, we have demonstrated that HHOT has many benefits for researchers working with tiled histopathology data. Future work may focus on using OT to direct dataset creation to optimize transferability, tune tile size and normalization hyper-parameters, and compare feature vectors learned from different models.

## Acknowledgments

R.G.K., R.M., D.A.M, and G.H were employed by Microsoft corporation while performing this work. A.Y. was employed by Microsoft corporation for a duration of this work. Part of this work has used computing resources at the NYU School of Medicine High Performance Computing Facility.

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
