# OpenReview forum: "Hierarchical Optimal Transport for Comparing Histopathology Datasets"
_MIDL.io/2022/Conference — MIDL 2022_

### Official Review · Reviewer_XJqD · 2022-01-24

**Confidence:** 4
**Preliminary Rating:** 2

**Summary:**

This paper addresses the need for comparing similarities of histopathology slides datasets for the purpose of transfer learning in computational pathology research. The claimed novelty is the definition of optimal transport (OT) distance between two slides by treating each slide as a distribution of tiles, and the definition of OT distance between two datasets by treating a dataset as a distribution of slides. The authors demonstrated the utility of such distance by computing the pair-wise distances between six histopathology slide datasets and evaluated 1) whether this distance aligns with basic ground truth (i.e., slides from the same type of cancer are close to each other and 2) whether the distances between datasets are indicative of transfer learning performance.

**Strengths:**

1. The paper is well written and easy to understand. This is particularly applaudable since the described method is a bit complex;
2. The proposed distance is theoretically novel and interesting.  I believe there is potential that the proposed distance can be useful in more difficult tasks and rare cancer datasets.

**Weaknesses:**

1. Fig 4 a) is confusing. I think the authors are missing the legend for LUAD (there is the plus sign in the figure that was not listed in the legend). The author should correct this figure. In addition, I think there is to be more discussion of this result. For example, why is it that the datasets sampled from LUSC seem to have larger distances between each other than with, say PAAD datasets? Is it because of the limitation of the proposed distance or because of the variations within LUSC datasets? I think it is important to discuss this with examples and visualizations.

2. I believe the utility of the proposed distance is inadequately demonstrated. The transfer learning study does not seem to support the authors' claim strongly (i.e. the linear regression correlation is somewhat weak).

**Deanonymize Review:**

no

**Final Rating After The Rebuttal:**

3: Borderline

**Justification Of The Final Rating:**

The authors corrected errors in the paper. However, I am still not convinced about the value of the proposed distance. If the distance can be used in other tasks, then some comparison with baselines in these tasks should be performed. Nonetheless I increase my rating to borderline.

**Paper Type:**

both

**Questions To Address In The Rebuttal:**

I hope the authors can address the weakness I proposed in the previous section:
1) To correct Fig 4. a) and provide a more detailed discussion of the results if possible;
2) Can the author use some other measure (qualitative or quantitative) to show the utility of the proposed distance?

**Special Issue:**

no

---

### Official Review · Reviewer_D7dZ · 2022-01-24

**Confidence:** 4
**Preliminary Rating:** 3
**Recommendation:** Poster

**Summary:**

The paper proposes a method to measure a distance between datasets. This distance can help selecting which dataset is more suitable for transfer learning for a particular task. The method is based on hierarchical generalization of optimal transport distance. The authors test their approach on histopathology data (TCGA dataset).

**Strengths:**

1. The paper proposes a method to select a more suitable dataset for transfer learning in histopathology. The hierarchical generalization of optimal transport distance is novel.
2. The paper is well written and is mostly easy to follow.
3. Identifying which dataset is more suitable for transfer learning can be useful to improve generalizability on under-profiled tasks.


**Weaknesses:**

1. The description of data used is missing, in particular: the number of slides per each task AND the number of patches extracted. What were the sizes of training/validation/test sets.
2. I noticed multiple problems with Figure 4. Please, see Detailed Comments section.


**Deanonymize Review:**

no

**Detailed Comments:**

The Figure 4 is supposed to visualize key results of a paper but it is quite confusing and has a poor visibility.

1. It is not mentioned what does each point on a Figure 4 (a) plots represents.

2. There are symbols on a figure that are NOT in label history (+).

3. There is a square in both label histories in Figure 4 (a), one that shows shape and the one that shows colours, I would advise to substitute one of them with a different symbol otherwise it is confusing.

4. The label history is missing the light green which is shown on a figure.

5. The analysis on results shown on Figure 4 (a) is missing.

5.1 In the case of STAD target it seems like pretraining on STAD gives the highest improvement in performance? Why is that?

5.2 In the case of LUSC it looks like pretraining on LUSC decreases performance? Why is that?

6. Some colors on Figure 4 (a) plots blend with the background.
7. Some points on Figure 4 (a) plots blend with each other, it would make sense to stretch the x-axis.




**Final Rating After The Rebuttal:**

4: Weak Accept

**Justification Of The Final Rating:**

The paper proposes a method to select a more suitable dataset for transfer learning in histopathology, based on hierarchical generalization of optimal transport distance. The authors addressed most of my concerns. I upgrade the rating to Weak Accept.

**Paper Type:**

both

**Questions To Address In The Rebuttal:**

1. Please add the detailed description of data: the number of slides per each task and the number of patches extracted. What were the sizes of training/validation/test sets.
2. Please address the points about Figure 4 (a) from Detailed Comments section.
3. What is the standard deviation of results across multiple training runs?


**Special Issue:**

no

---

### Official Review · Reviewer_hVUB · 2022-01-26

**Confidence:** 5
**Preliminary Rating:** 4
**Recommendation:** Poster

**Summary:**

In this paper, they proposed a method to compare similarity of histopathology datasets which can be used in transfer learning techniques where there is a large labeled dataset and a small target dataset. Their method was based on optimal transport method which performed hierarchically on the tiles for all pairs of slides, and then the slides between datasets. They tested their method on H&E stained slides of six datasets of TCGA from different cancer types and sowed that their method outperforms a baseline method in prediction of cancer type, and also can predict the difficulty of transferability.

**Strengths:**

- Tiling over whole slide images (using small patches), which is applied before most deep learning models, is considered explicitly in this method to adopt the optimal transport to histopathology slides.
- The method facilitates using a publicly available dataset from different tissues for pre-training a model to apply to a small (local) dataset.
- The discussion and the results deal with the various aspects of the method visually and quantitatively.
- The concept is clear, and English is good


**Weaknesses:**

- The abstract and Introduction talked about deep learning; however, in the paper, the method was tested by a single layer perceptron which can cause doubt about the generalizability.
- The method is well discussed and concluded; however, as it is applied to histopathology images, it is better to briefly discuss pathology. For example, If two datasets have the most similarity, do the pathologists also confirm this.

**Deanonymize Review:**

no

**Final Rating After The Rebuttal:**

4: Weak Accept

**Justification Of The Final Rating:**

The authors replied most of the questions and comments very well. Although there is still lack of pathological explanation (like morphology of similar cancers), the paper is good enough to be accepted in MIDL 2022.

**Paper Type:**

both

**Questions To Address In The Rebuttal:**

- Due to the aim of using histopathology images, different pixel spacing and patch sizes are used. Would you please explain the effect of these parameters on your method? Have you chosen them empirically, or are these optimum?
- Would you please explain that the definition of transferability depends on the classifier (perceptron) and features you selected or not? In other words, if you choose a more complex model, the AUC of DT is more close to optimal and will be increased, so the transferability of every dataset will be decreased.
- Would you please explain (if possible) that the similarity between four datasets is pathologically justifiable?
- The HHOT distance is calculated in the training stage just once;  is reducing the calculation time considered a significant advantage of your method? Would you please mention the processing specification of the system you used to achieve the results in Figure 4(c), Y-axis?

**Special Issue:**

no

---

### Meta-Review · Area_Chair_i8SH · 2022-02-20

**Recommendation:** Accept (Poster)
**Confidence:** 4

**Metareview:**

All reviewers agree that the paper has value and proposed a fairly novel idea.
This method is solely applied to digital pathology images and deserves future work, but it is interesting for the MIDL community.

---

### Decision · Program_Chairs · 2022-02-28

Accept